# Madurastatins with Imidazolidinone Rings: Natural Products or Side-Reaction Products from Extraction Solvents?

**DOI:** 10.3390/ijms25010301

**Published:** 2023-12-25

**Authors:** Mercedes Pérez-Bonilla, Marina Sánchez-Hidalgo, Ignacio González, Daniel Oves-Costales, Jesús Martín, José Murillo-Alba, José R. Tormo, Ahreum Cho, Soo-Young Byun, Joo-Hwan No, David Shum, Jean-Robert Ioset, Olga Genilloud, Fernando Reyes

**Affiliations:** 1Fundación MEDINA, Avda. del Conocimiento 34, 18016 Granada, Spain; marina.sanchez@medinaandalucia.es (M.S.-H.); ignacio.gonzalez@medinaandalucia.es (I.G.); daniel.oves@medinaandalucia.es (D.O.-C.); jesus.martin@medinaandalucia.es (J.M.); jose.murillo@medinaandalucia.es (J.M.-A.); ruben.tormo@medinaandalucia.es (J.R.T.); olga.genilloud@medinaandalucia.es (O.G.); 2Institut Pasteur Korea, 16, Daewangpangyo-ro 712 beon-gil, Bundang-gu, Seongnam-si 13488, Gyeonggi-do, Republic of Korea; ahreum.cho@ip-korea.org (A.C.); sooyoung.byun@ip-korea.org (S.-Y.B.); joohwan.no@ip-korea.org (J.-H.N.); david.shum@ip-korea.org (D.S.); 3Drugs for Neglected Diseases Initiative, 15 Chemin Camille-Vidart, 1202 Geneva, Switzerland; jrioset@dndi.org

**Keywords:** madurastatin, neglected tropical diseases, *Leishmania donovani*, *Trypanosoma cruzi*

## Abstract

Madurastatins are a group of pentapeptides containing an oxazoline moiety, and, in a few cases, an imidazolidinone ring as an additional structural feature. In our search for new potential antiparasitic metabolites from natural sources, we studied the acetone extracts from a culture of *Actinomadura* sp. CA-135719. The LC/HRMS analysis of this extract identified the presence of the known madurastatins C1 (**1**), D1 (**4**), and D2 (**5**) together with additional members of the family that were identified as the new madurastatins H2 (**2**) and 33-*epi*-D1 (**3**) after isolation and spectroscopic analysis. The planar structures of the new compounds were established by HRMS, ESI-qTOF-MS/MS, and 1D and 2D NMR data, and their absolute configuration was proposed using Marfey’s and bioinformatic analyses of the biosynthetic gene cluster (BGC). A revision of the absolute configuration of madurastatins D1 and D2 is proposed. Additionally, madurastatins containing imidazolidinone rings are proved to be artifacts originating during acetone extraction of the bacterial cultures.

## 1. Introduction

Since the first report on the isolation of madurastatins A1–A3 and B1–B2 from an *Actinomadura madurae* IFM 0745 strain [1], the structures of 24 different madurastatins (Figure 1) have been reported [2,3,4,5,6,7]. The original structures of madurastatins containing an aziridine ring were recently revised and reassigned as pentapeptides containing an unusual salicylate-capped *N*-terminal 2-(2-hydroxyphenyl)-oxazoline in place of the originally postulated aziridine [8]. Additionally, among the most recent madurastatins reported, the presence of an unusual 4-imidazolidinone ring was observed in madurastatins D1 and D2 [4], D3 and D4 [6], and E1 [5]. The absolute configuration of madurastatins was established through Marfey’s analysis, DP4+ calculation, phylogenetic domain analyses of the putative biosynthetic gene cluster, and total synthesis. All these analyses revealed the presence of D-serine, *N*α-methyl-*N*δ-hydroxy-L-ornithine, and *N*δ-hydroxy-L-ornithine in all madurastatins described [5,6,8] except for madurastatins D1 and D2, isolated together with *ent*-madurastatin C1, in which the constituent amino acids were reported to have the opposite absolute configuration [4].

Leishmaniasis and American trypanosomiasis (Chagas disease) are two neglected tropical diseases (NTDs), caused by the parasites *Leishmania* spp. and *Trypanosoma cruzi*, respectively, that lead to thousands of deaths worldwide every year. These diseases are also emerging as a health problem in developed countries. The lack of field-adapted treatments, due to limitations including suboptimal efficacy, drug resistance, and adverse effects, poses serious threats to human health, and new therapeutic solutions are required [9,10]. Natural products are unique sources of chemical diversity and have historically constituted rich sources of new bioactive compounds, with many drugs originated from natural products being used today in clinical practice. Thus, microbial natural products remain a unique and untapped source for the identification of novel bioactive compounds [11].

As part of an ongoing partnership between Fundación MEDINA, Institut Pasteur Korea (IPK), and Drugs for Neglected Diseases initiative (DNDi) to discover natural products active against *T. cruzi* and *L. donovani*, a subset of the MEDINA natural products extracts libraries was tested against both parasites in the IPK phenotypic screening platform. LC-HRMS analysis of the active acetone extract from a culture of *Actinomadura* sp. CA-135719 in FPY-6 medium revealed, together with madurastatin C1, the presence of other compounds with molecular formulae corresponding to potentially new madurastatins that prompted us to a more detailed investigation of their structures. 

## 2. Results and Discussion

### 2.1. LC-UV-HRMS Analysis of the Extract from Actinomadura sp. CA-135719 and Identification of Components

The acetone extract from a culture in FPY-6 medium of *Actinomadura* sp. CA-135719 displayed the LC-UV chromatographic profile at 210 nm shown in Figure 2. The major peaks (**1** and **5**) together with the minor metabolites (**2**–**4**) were isolated, and their structures were established by HRMS and 1D and 2D NMR analysis. The major components of the extracts were identified as madurastatins C1 (**1**) and D2 (**5**), whereas molecular formulae of C_29_H_43_N_7_O_10_ (**2**) and C_28_H_39_N_7_O_9_ (**3** and **4**) were assigned to the minor components of the extract. 

### 2.2. Extraction, Isolation, and Structural Elucidation

A culture (3 L) from *Actinomadura* sp. CA-135719 fermented in FPY-6 medium for 7 days was extracted with an equal volume of acetone and subsequently fractionated by column chromatography using SP207ss resin followed by reversed-phase preparative HPLC using an XBridge C18 column, which led to the isolation of madurastatin C1 (**1**) (Figure 1), together with the new madurastatins H2 (**2**) and 33-*epi*-D1 (**3**) and the known madurastatins D1 (**4**) and D2 (**5**) (Figure 3). 

Compound **2** was obtained as an optically active light-yellow sirup (${\left[\alpha \right]}_{D}^{25}$ − 5.2, MeOH). The (+)-ESI-qTOF analysis showed a protonated adduct at *m*/*z* 650.3145 [M + H]^+^, from which the molecular formula of C_29_H_43_N_7_O_10_ was deduced. The IR spectrum showed characteristic absorptions of hydroxy and NH (3285 cm^−1^), C-H (2897, 2837 cm^−1^), carbonyl (1708 cm^−1^) and olefinic groups (1541, 1482 cm^−1^). The planar structure of **2** was confirmed by 1D and 2D NMR spectra (Figure 4) and MS/MS fragmentation (Figure 5). Its ^1^H NMR spectrum displayed signals attributable to a pentapeptide, similar to those found in related madurastatins [4,5,6,8]. NMR spectra displayed signals corresponding to a salicylate unit followed by serine, glycine, β-alanine, *N*α-methyl-*N*δ-hydroxy-ornithine, and a cyclic *N*δ-hydroxy-ornithine moiety (Table 1). Unlike madurastatin C1 (**1**), compound **2** showed a serine unit instead of an oxazoline moiety, inferred by the upfield shift of ^13^C and ^1^H NMR signals at C-9 and the upfield shift of C-10 and C-11. The structure was confirmed by the characteristic fragment at *m*/*z* 208.0601 observed in MS^2^-data (Figure 5). An additional difference compared to madurastatin C1 was the presence of a tetrasubstituted 4-imidazolidinone ring as in madurastatins D1 and D2 [4]. Two methyl groups at δ_H_ 1.32 and 1.50 displayed HMBC correlations to each other’s carbon and with a quaternary carbon at δ_C_ 80.2 (Figure 4), indicating the connectivity and the presence of the imidazolidinone ring. Additionally, the fragment at *m*/*z* 171.1129 also confirms this assumption (Figure 5). Compound **2** was named madurastatin H2 based on its structural similarity with madurastatin H1.

The molecular formula of C_28_H_39_N_7_O_9_ assigned to compounds **3** and **4** by HRMS analysis was previously found in madurastatin D1 [4]. Analysis of the 1D and 2D NMR data revealed both compounds to have the same planar structure as madurastatin D1, although noticeable differences in the ^1^H and ^13^C chemical shifts of C-22, C-23, C-26, C-32, C-33, and C-34 between both compounds were observed (Table 2). The NMR data of compound **4** are identical to those published for madurastatin D1 (Figure 3). The presence of a methyl group at δ_H_ 1.12 and δ_C_ 15.1 and the downfield chemical shift of H-33 to δ_H_ 4.42 in compound **3** suggest a different configuration at C-33, which leads to the aforementioned differences in chemical shifts at the surrounding positions. Thus, compound **3** was confirmed to be the 33-epimer of compound **4** (Figure 3). 

Once the planar structure of the compounds was established, Marfey’s analysis was used to determine the absolute configuration of the constituent amino acids [12]. Acid hydrolysis of **1**–**4** was followed by LC-MS analysis of the hydrolysates after derivatization with *N*-(2,4-dinitro-5-fluorophenyl)-D/L-valinamide (D/L-FDVA, Marfey’s reagent) and comparison with the retention times and mass spectra obtained for standards. Since *N*α-methyl-L-Orn was not available, its absolute configuration determination was based on the D/L elution order described in bibliography [6]. Whereas the elution order of the doubly derivatized Orn-L-FDVA was determined to be D→L, this elution order was inverted in the case of *N*α-methyl-Orn-L-FDVA, being L→D [6]. Partial racemization was observed in all double ornithine adducts of the compound. Marfey’s analyses confirmed the presence of D-Ser, *N*δ-hydroxy-L-Orn, and *N*α-methyl-*N*δ-hydroxy-L-Orn in compounds **1**–**4** (Appendix A). Thus, the absolute configuration of madurastatins was established as 9*R*, 23*S*, 26*S*. 

To our knowledge, all madurastatins described contain D-serine, *N*α-methyl-*Nδ*-hydroxy-L-Orn, and *N*δ-hydroxy-L-Orn as constituent amino acids except those reported by Yan and coworkers, who reported the isolation of *ent*-madurastatin C1, mainly based on a negative value of its specific rotation [4]. The Marfey’s analysis carried out by these authors [4] to confirm their stereochemical proposal was limited to comparing the retention times of D- and L-Ser standards derivatized with L-FDLA, where the difference found in retention time between the L- and D-Ser Mosher derivatives was less than 0.1 min. Poor chromatographic resolution of the D/L-Ser pair derivatized with L-FDLA using formic acid-based solvent systems was previously reported [13]. A confirmation of the absolute configuration of other constituent amino acids of the molecule, together with the development of an analytical method with a better chromatographic resolution of the D/L-Ser Mosher derivatives pair, to unequivocally establish the absolute configuration of the Ser residue and hence that of the claimed *ent*-madurastatin C1 would have been desired. The absolute configuration of madurastatins D1 and D2, isolated in the same work, was assumed to be the same as that of *ent*-madurastatin C1. 

As mentioned above, compounds **3** and **4** differ in their absolute configuration at C-33. Compound **4** was demonstrated to have the same NMR spectroscopic data as madurastatin D1 and hence the same relative configuration. However, based on a common biosynthetic origin with madurastatins C1 and H2 isolated from our CA-135719 strain, we propose that **4** has an opposite absolute configuration in all its chiral centers (9*R*, 23*S*, 26S, 33*S*) to that reported previously for madurastatin D1. The positive specific rotation measured for both compounds confirms that the absolute configuration initially proposed for madurastatin D1 is most probably wrong [4].

### 2.3. In Silico Analysis of the Madurastatin Biosynthetic Gene Cluster

To confirm the difference in the absolute configuration between **4** and that previously reported for madurastatin D1, the genome of the strain CA-135719 was sequenced and assembled using a combination of PacBio and Illumina NovaSeq. A complete circular genome of ~9.5 Mb was obtained, and four identical rRNA 16S gene sequences showing a 98.80% similarity with *Actinomadura darangshiensis* DSLS-70T were identified. AntiSMASH analysis predicted 22 putative BGCs, including a 75 Kb cluster similar to the clusters encoding madurastatins C1, D1, and D2 (*mad*) [4] and madurastatins A1, A2, E1, F, and G1 (*rene*) [5] (Appendix A). The cluster contained 61 genes (*mds1*-*mds61*), including four NRPS (*mds12*, *mds21*, *mds24*, and *mds37*), an L-ornithine-N-(5)-monooxygenase (*mds39*) involved in *N*δ-hydroxy-ornithine biosynthesis, an aspartate 1-decarboxylase (*mds23*) involved in β-Ala biosynthesis, and a salicylate synthase (*mds36*) responsible of salicylic acid biosynthesis (Appendix A, Figure 6a). It has been proposed that the biosynthesis of madurastatins begins with the loading of the salicylate moiety to the thiolation domain of Mds21 by Mds24 (Figure 6b) [5]. The heterocyclization domain (C1) of Mds21 generates a phenyloxazoline ring from the condensation of salicylate and D-Ser (Figure 6b). Since no epimerases are present in the cluster, the conversion of L-Ser to D-Ser remains uncertain [5]. Then, Mds37 incorporates glycine, β-alanine, and *N*δ-hydroxy-acetyl-L-ornithine, according to the predictions of the first three NRPS modules (Figure 6b). The last module of Mds37, as in the case of ReneL [5], lacks the adenylation (A) and thioesterase (TE) domains that would incorporate *N*δ-hydroxy-L-ornithine and release the NRP, respectively. While Yan et al. [4] described the presence of a ^D^C_L_ domain and an ornithine-specific A domain in the last module of the homologous NRPS Mad30, a reanalysis of the mad cluster shows that the last module shares the same organization as ReneL and Mds37 (presence of a ^L^C_L_ domain and absence of an A domain) (Appendix A), so no D-amino acids may be incorporated in that position. Moreover, except for the D-Ser from the phenyloxazoline ring, all the incorporated amino acids should be L-configured since no epimerization (E) or dual E/C domains are present in the NRPS (Figure 6b and Appendix A). A trans-acting A domain (Mds12 or Mds48) may activate the *N*δ-hydroxy-L-ornithine incorporated in the last module of Mds37, similar to the proposed biosynthesis of madurastatin by the *rene* cluster [5]. This should also occur in the *mad* cluster [4]. The chain release might be catalyzed by the esterase Mds41, although a spontaneous nonenzymatic hydrolysis cannot be excluded [5].

The biosynthesis of madurastatin C1 and the rest of the compounds isolated from CA-135719 has been clearly established, and BGC analysis confirmed the absolute configuration determined by Marfey’s analysis. Similarities found between the *mds* and the *mad* and *rene* clusters allow us to conclude that the absolute configuration previously proposed for madurastatins D1 and D2 [4] is probably wrong and that the existence of *ent*-madurastatin C is questionable and might be explained due either to an error when measuring the specific rotation of the compound or to the specific rotation having been measured with a metal-chelated species of the molecule.

Finally, due to the absence in the *mds* cluster of genes related to the biosynthesis of the 4-imidazolidinone ring present in **2**–**5** and the fact that two different madurastatins D1 having opposite configuration at C-33 were isolated, we postulate that madurastatins **2**–**5** might be a consequence of the reaction of the natural madurastatins with the acetone used in the extraction. 

### 2.4. Effect of the Solvent Used in the Extraction of Madurastatins

To verify this hypothesis, small-scale fermentations of strain CA-135719 in FPY-6 medium for 7 days were extracted with an equal volume of five solvents, acetonitrile, n-butanol, ethyl acetate, methanol, and acetone, in triplicate. The extracts were analyzed by LC-HRMS, and ion extraction of each madurastatin was performed (Table 3). Whereas madurastatins C1 and H1 were detected in all extracts as major and minor compounds, respectively, madurastatins H2, 33-*epi*-D1, D1, and D2 were only detected at very low levels in acetone extracts. Consequently, the formation of madurastatins H2 and D2 can be explained as originating from the reaction of madurastatins H1 and C1, respectively, with acetone (Appendix A). Similarly, the isolation of the two madurastatin D1 epimers (**3** and **4**) might be a result of a nonenzymatic addition of acetaldehyde to madurastatin C1 (Appendix A). 

Although this side reaction extensively occurs on the *N*-termini of peptides during dimethyl labeling using formaldehyde and significantly reduces the quality of proteomic analysis, it is frequently overlooked because of the limitations of current database search engines on the identification of unknown modifications [14]. Other byproducts containing the imidazolidinone ring were observed in the synthesis of the precursors of discarines C and D and myrianthine A [15] and in nummularin G [16], sativanine B [17], and cambodines A, B, D, E, and F [18]. Additionally, several vancomycin antibiotics also undergo spontaneous chemical modifications when kept at room temperature at physiological pH in aqueous solutions containing traces of formaldehyde or acetaldehyde. By using tandem mass spectrometry, the modification was unambiguously identified as a 4-imidazolidinone moiety at the *N*-terminus [19]. The formation of 4-imidazolidinone moieties as formaldehyde adducts is not unprecedented and is indicated in several studies [20,21,22,23,24,25].

### 2.5. Obtainment of Madurastatin D2 from Madurastatin C1 Using Acetone

To confirm that madurastatin D2 can be originated from madurastatin C1 using acetone as solvent, the latter (Figure 7a) was incubated in acetone. As it can be observed after the incubation with acetone, the LC-MS profile revealed not only the presence of madurastatin C1 but also the presence of madurastatin D2 (Figure 7b). Thus, acetone can be incorporated into madurastatin C1 with the subsequent formation of the imidazolidinone ring to give madurastatin D2 (Appendix A).

The extraction of the CA-135719 cultures with different solvents (Table 3) as well as the incubation of madurastatin C1 with acetone demonstrate that madurastatin D2 is an artifact produced by acetone. On the other hand, the origin of acetaldehyde to form compounds **3** and **4** is uncertain. Nonetheless, the absence of these compounds in extracts obtained in other solvents than acetone, together with the absence of genes in the mds cluster that explain the biosynthesis of the 4-imidazolidinone ring, together with the isolation of both epimers at C-33 of madurastatin D1 indicate that these compounds might not be the product of an enzymatic reaction.

### 2.6. Antiparasitic Activity

All compounds isolated were tested against *Trypanosoma cruzi* and *Leishmania donovani*, using the liquid microdilution method. The IC_50_ values against both parasites are indicated in Table 4. Madurastatins were not active against *L. donovani* at the highest concentration tested, showing IC_50_ values higher than 50 µM. Against *Trypanosoma cruzi*, madurastatin activities ranged from moderate to low, although the compounds also displayed some cytotoxicity. Madurastatin D2 displayed the lowest IC_50_ (8.93 µM) for the inhibition normalized by parasite number. 

## 3. Materials and Methods

### 3.1. General Experimental Procedures

Optical rotations were measured using a Jasco P-2000 polarimeter (JASCO Corporation, Tokyo, Japan). UV spectra were obtained with an Agilent 1100 DAD (Agilent Technologies, Santa Clara, CA, USA). IR spectra were recorded on a JASCO FT/IR-4100 spectrometer (JASCO Corporation, Tokyo, Japan) equipped with a PIKE MIRacle^TM^ single-reflection ATR accessory. NMR spectra were recorded on a Bruker Avance III spectrometer (500 and 125 MHz for ^1^H and ^13^C NMR, respectively) equipped with a 1.7 mm TCI MicroCryoProbe^TM^ (Bruker Biospin, Falländen, Switzerland). Chemical shifts were reported in ppm using the signals of the residual solvent as internal reference (δ_H_ 2.50 and δ_C_ 39.51 for DMSO-*d*_6_). LC-MS and LC-HRMS analyses were performed as described previously [26].

### 3.2. Taxonomic Identification of the Producing Microorganism

The producing strain CA-135719 was obtained from a rhizosphere sample associated to *Veronica anagallis-aquatica*, collected in the Baviaanskloof mountains in the Eastern Cape of South Africa. This rhizosphere soil was air-dried, heated at 100 °C for 1 h, and then suspended in sterile water. The soil suspension was serially diluted, plated on selective isolation media, and incubated at 28 °C for at least 6 weeks. The strain was isolated from an NZ-amine-based agar medium containing nalidixic acid (20 µg/mL). The colony was purified on yeast extract, malt extract, glucose medium (ISP2) and preserved as frozen agar plugs in 10% glycerol.

The partial 16S rRNA gene sequence (1481 nucleotides) of strain CA-135719 was compared with those deposited in public databases and the EzBiocloud server (https://www.ezbiocloud.net/, accessed on 8 November 2022; [27]). The strain exhibited the highest similarity (98.80%) with *Actinomadura darangshiensis* DSLS-70T (FN646682), using EzBiocloud and GenBank sequence similarity searches and homology analysis.

The morphological and 16S rRNA gene sequence data were indicative that strain CA-135719 was representative of members of the genus *Actinomadura* and the strain was referred to as *Actinomadura* sp. CA-135719.

### 3.3. Fermentation of the Producing Microorganism

A 3 L fermentation of the producing microorganism was generated as follows: the first seed culture of the strain CA-135719 was prepared by inoculating 10 mL of seed medium, which consists of soluble starch (20 g/L), dextrose (10 g/L), NZ amine EKC (Sigma) (5 g/L), Difco beef extract (3 g/L), Bacto peptone (5 g/L), yeast extract (5 g/L), and CaCO_3_ (1 g/L), adjusted to pH 7.0 with NaOH before addition of CaCO_3_, in a 40 mL tube with 0.5 mL of a frozen inoculum stock of the producing strain and incubation of the tube at 28 °C with shaking at 220 rpm for about 48 h. A second seed culture was prepared by inoculating 50 mL of seed medium in two 250 mL flasks with 2.5 mL of the first seed. The content of both flasks was then mixed, and a 5% aliquot of the mixture was used to inoculate twenty-four 500 mL flasks containing 125 mL of the production medium consisting of fructose 10 g/L, glucose 10 g/L, bacto peptone 2 g/L, bacto yeast extract 5 g/L, NZ-Amine E (EKC) 5 g/L, amicase 5 g/L, deionized water 1000 mL, trace elements 1 mL, pH 7.0. Trace elements: FeSO_4_·7H_2_O 500 mg/L, ZnSO_4_·7H_2_O 500 mg/L, MnSO_4_·H_2_O 100 mg/L, CuSO_4_·5H_2_O 50 mg/L, CoCl_2_·6H_2_O 50 mg/L. The flasks were incubated at 28 °C for 7 days in a rotary shaker at 220 rpm and 70% humidity before harvesting.

### 3.4. Extraction and Isolation

After 7 days of fermentation, acetone was added to the fermentation flasks (1:1, 125 mL), and they were shaken in a Kühner at 220 rpm for 2 h. After that, the mixture was centrifuged (10 min, 8500 rpm) and filtered under vacuum, and the pellet was discarded. Acetone was evaporated under a nitrogen stream until the original volume of fermentation (3 L) to obtain an aqueous residue.

The aqueous residue was loaded onto an SP207ss resin column (76 g, 32 × 100 mm) and eluted with an H_2_O−acetone increasing gradient (90/10 to 0/100 for 50 min, and 100/0 for 10 min, 10 mL/min, 15 mL/fraction) in a Teledyne CombiFlash RF apparatus to give 40 fractions. Fraction 22 was subjected to preparative reversed-phase HPLC (XBridge C_18_, 19 × 250 mm, 5 µm, 14 mL/min, UV detection at 210 and 280 nm, 7 mL/fraction) using H_2_O (solvent A) and CH_3_CN (solvent B). Elution was carried out using isocratic conditions of 5% B for 1 min and then a linear gradient from 5% to 50% B for 34 min, followed by a linear gradient from 50% to 100% in 1 min and held at 100% B for 7 min, yielding 80 fractions. 

Subfraction 38 was further purified by semipreparative reversed-phase HPLC (XBridge C_18_, 10 × 150 mm, 5 µm, 3.6 mL/min, UV detection at 210 and 280 nm, 1.8 mL/fraction) using H_2_O (solvent A) and CH_3_CN (solvent B). Elution was carried out using isocratic conditions of 10% B for 3 min, followed by a linear gradient from 10% to 35% for 31 min, then a linear gradient from 35% to 100% in 2 min and held at 100% for 7 min to yield madurastatin C1 (**1**) (t_R_ 24.0 min, 11.7 mg) and compound **2** (t_R_ 26.0 min, 2.7 mg).

Subfraction 44 was further purified by semipreparative reversed-phase HPLC (XBridge C_18_, 10 × 150 mm, 5 µm, 3.6 mL/min, UV detection at 210 and 280 nm, 1.8 mL/fraction) using H_2_O (solvent A) and CH_3_CN (solvent B). Elution was carried out using isocratic conditions of 10% B for 1 min, followed by a linear gradient from 10% to 35% for 35 min, then a linear gradient from 35% to 100% in 1 min and held at 100% for 6 min to yield 33-*epi*-madurastatin D1 (**3**) (t_R_ 23.0 min, 1.4 mg).

Subfraction 45 was further purified by semipreparative reversed-phase HPLC (XBridge C_18_, 10 × 150 mm, 5 µm, 3.6 mL/min, UV detection at 210 and 280 nm, 1.8 mL/fraction) using H_2_O (solvent A) and CH_3_CN (solvent B). Elution was carried out using isocratic conditions of 10% B for 1 min, followed by a linear gradient from 10% to 35% for 35 min, then a linear gradient from 35% to 100% in 1 min and held at 100% for 6 min to yield madurastatin D1 (**4**) (t_R_ 25.0 min, 0.9 mg).

Subfraction 50 was further purified by semipreparative reversed-phase HPLC (XBridge C_18_, 10 × 150 mm, 5 µm, 3.6 mL/min, UV detection at 210 and 280 nm, 1.8 mL/fraction) using H_2_O (solvent A) and CH_3_CN (solvent B). Elution was carried out using isocratic conditions of 10% B for 1 min, followed by a linear gradient from 10% to 35% for 35 min, then a linear gradient from 35% to 100% in 1 min and held at 100% for 6 min to yield madurastatin D2 (**5**) (t_R_ 31.5 min, 3.0 mg).

Madurastatin C1 (**1**): light-yellow sirup; ${\left[\alpha \right]}_{D}^{25}$ + 5.4 (*c* 2.82, MeOH); UV (DAD) λ_max_ 210, 248, 305 nm; ^1^H (500 MHz, CD_3_OD) *δ* 6.97 (1H, *d*, *J* = 8.3 Hz, H-2), 7.42 (1H, *ddd*, *J* = 8.6, 7.2, 1.5 Hz, H-3), 6.91 (1H, *dd*, *J* = 8.3, 7.4 Hz, H-4), 7.70 (1H, *dd*, *J* = 7.9, 1.2 Hz, H-5), 5.05 (1H, *dd*, *J* = 10.5, 8.3 Hz, H-9), 4.69 (1H, *dd*, *J* = 10.5, 8.3 Hz, H-10a), 4.61 (1H, *dd*, *J* = 9.4, 8.3 Hz, H-10b), 3.94 (1H, *d*, *J* = 16.7 Hz, H-13a), 3.85 (1H, *d*, *J* = 16.7 Hz, H-13b), 3.49–3.44 (2H, *m*, H-16), 2.75–2.65 (2H, *m*, H-17), 3.68–3.61 (1H, *m*, H-20a), 3.59–3.52 (1H, *m*, H-20b), 1.75–1.69 (2H, *m*, H-21), 1.68–1.61 (1H, *m*, H-22a), 1.61–1.55 (1H, *m*, H-22b), 3.07 (1H, *t*, *J* = 6.4 Hz, H-23), 4.50 (1H, *q*, *J* = 5.4 Hz, H-26), 3.65–3.60 (1H, *m*, H-29a), 3.60–3.55 (1H, *m*, H-29b), 2.04–1.95 (2H, *m*, H-30), 2.10–2.05 (1H, *m*, H-31a), 1.80 (1H, *qd*, *J* = 12.2, 3.5 Hz, H-31b), 2.32 (3H, s, H-32); ^13^C (CD_3_OD, obtained from HSQC) *δ* 117.5 (C-2), 134.9 (C-3), 119.8 (C-4), 129.3 (C-5), 69.3 (C-9), 70.4 (C-10), 43.4 (C-13), 36.2 (C-16), 32.7 (C-17), 48.2 (C-20), 23.5 (C-21), 31.1 (C-22), 64.6 (C-23), 51.1 (C-26), 52.4 (C-29), 21.6 (C-30), 28.6 (C-31), 34.3 (C-32); (+)-ESI-qTOF MS *m*/*z* 592.2761 [M + H]^+^ (calculated for C_26_H_38_N_7_O_9_^+^, 592.2726). 

Madurastatin H2 (**2**): light-yellow sirup; ${\left[\alpha \right]}_{D}^{25}$ − 5.2 (*c* 2.5, MeOH); UV (DAD) λ_max_ 210, 248, 305 nm; IR (ATR) ν_max_ 3285, 2897, 2837, 1708, 1541, 1482, 1359, 1301, 1193, 1027 cm^−1^; for ^1^H and ^13^C NMR data, see Table 1; (+)-ESI-qTOF MS *m*/*z* 650.3149 [M + H]^+^ (calculated for C_29_H_44_N_7_O_10_^+^, 650.3144). 

33-*epi*-madurastatin D1 (**3**): light-yellow sirup; ${\left[\alpha \right]}_{D}^{25}$ + 40.7 (*c* 0.75, MeOH); UV (DAD) λ_max_ 215, 245, 305 nm; for ^1^H and ^13^C NMR data, see Appendix A; (+)-ESI-qTOF MS *m*/*z* 618.2879 [M + H]^+^ (calculated for C_28_H_40_N_7_O_9_^+^, 618.2882). 

Madurastatin D1 (**4**): light-yellow sirup; ${\left[\alpha \right]}_{D}^{25}$ + 112.2 (*c* 0.15, MeOH); UV (DAD) λ_max_ 216, 248, 307 nm; for ^1^H and ^13^C NMR data, see Appendix A; (+)-ESI-qTOF MS *m*/*z* 618.2885 [M + H]^+^ (calculated for C_28_H_40_N_7_O_9_^+^, 618.2882). 

Madurastatin D2 (**5**): light-yellow sirup; ${\left[\alpha \right]}_{D}^{25}$ + 44.1 (*c* 1.02, MeOH); UV (DAD) λ_max_ 222, 245, 305 nm; ^1^H (500 MHz, CD_3_OD) *δ* 6.97 (1H, *brd*, *J* = 8.3 Hz, H-2), 7.42 (1H, *ddd*, *J* = 8.8, 7.3, 1.7 Hz, H-3), 6.90 (1H, *ddd*, *J* = 8.3, 7.5, 1.0 Hz, H-4), 7.69 (1H, *dd*, *J* = 7.9, 1.5 Hz, H-5), 5.05 (1H, *dd*, *J* = 10.6, 8.2 Hz, H-9), 4.68 (1H, *dd*, *J* = 10.6, 8.6 Hz, H-10a), 4.60 (1H, *t*, *J* = 8.3 Hz, H-10b), 3.95 (1H, *m*, H-13a), 3.83 (1H, *d*, *J* = 16.7 Hz, H-13b), 3.47 (1H, *t*, *J* = 6.5 Hz, H-16), 2.70 (2H, *m*, H-17), 3.58 (2H, *m*, H-20), 1.78 (1H, *m*, H-21a), 1.56 (1H, m, H-21b), 1.68 (1H, *m*, H-22a), 1.61 (1H, *m*, H-22b), 3.16 (1H, *m*, H-23), 3.95 (1H, *m*, H-26), 3.70 (1H, *m*, H-29a), 3.56 (1H, *m*, H-29b), 1.98 (2H, *m*, H-30), 2.44 (1H, *m*, H-31a), 1.88 (1H, *m*, H-31b), 2.33 (3H, *s*, H-32), 1.40 (3H, *s*, H-34), 1.22 (3H, *s*, H-35); ^13^C (CD_3_OD, obtained from HSQC) *δ* 117.4 (C-2), 134.8 (C-3), 119.8 (C-4), 129.3 (C-5), 69.2 (C-9), 70.4 (C-10), 43.4 (C-13), 36.2 (C-16), 32.7 (C-17), 48.7 (C-20), 22.0 (C-21), 26.2 (C-22), 63.8 (C-23), 53.6 (C-26), 52.2 (C-29), 22.1 (C-30), 26.7 (C-31), 32.6 (C-32), 25.4 (C-34), 20.4 (C-35); (+)-ESI-qTOF MS *m*/*z* 632.3056 [M + H]^+^ (calculated for C_29_H_42_N_7_O_9_^+^, 632.3039). 

### 3.5. Marfey’s Analysis of Compounds ***1***–***4***

A sample of compound **2** (700 μg) was dissolved in 0.7 mL of 6 N HCl and heated at 110 °C for 16 h. The crude hydrolysate was evaporated to dryness under a N_2_ stream, and the residue was dissolved in 100 μL of water. This solution was divided into two 50 μL aliquots. To each aliquot of the hydrolysate and to an aliquot (50 μL) of a 50 mM solution of each amino acid (D and L), 20 μL of 1 M NaHCO_3_ solution and a 1% (*w*/*v*) solution (100 μL) of D or L-FDVA (Marfey’s reagent, *N*-(2,4-dinitro-5-fluorophenyl)-L-valinamide) was added. The reaction mixture was incubated at 40 °C for 60 min. After this time, the reaction was quenched by the addition of 10 μL of 1 N HCl, and the crude mixture was diluted with 200 μL of acetonitrile and analyzed by LC/MS on an Agilent 1260 Infinity II single-quadrupole LC/MS instrument. Separations were carried out on an Atlantis T3 column (4.6 × 100 mm, 5 μm) maintained at 40 °C. A mixture of two solvents, A (10% acetonitrile, 90% water) and B (90% acetonitrile, 10% water), both containing 1.3 mM trifluoroacetic acid and 1.3 mM ammonium formate, was used as the mobile phase under a linear gradient elution mode (isocratic 20% B for 2 min, 20−45% B in 27 min, isocratic 45% B for 5 min) at a flow rate of 1 mL/min.

Retention times (min) for the derivatized (D-FDVA) amino acid standards under the reported conditions were as follows when Marfey’s analysis was performed for compound **2**: D-Ser: 6.86, L-Ser: 8.06, single adducts of D-Orn: 3.56 and 5.33, double adduct of D-Orn: 27.65, single adducts of L-Orn: 3.18 and 5.32, double adduct of L-Orn: 26.14 (Appendix A). Retention times (min) for the observed peaks in the HPLC trace of the D-FDVA-derivatized hydrolysis product of compound **2** were as follows: D-Ser: 6.79, Gly: 10.49, β-Ala: 13.44, double adduct of L-Orn: 26.16, and double adduct of *N*α-methyl-L-Orn: 31.04 (Appendix A). 

Samples of compounds **1**, **3**, and **4** (400, 400, and 200 μg, respectively) were dissolved in 6 N HCl (0.8, 0.8, and 0.4 mL, respectively) and heated at 160 °C for 7 h. The crude hydrolysates were evaporated to dryness under a N_2_ stream, and the residues were dissolved in 200 μL of water. To these solutions, 100 μL of 1 M NaHCO_3_ solution and a 1% (*w*/*v*) solution (100 μL) of D-FDVA were added. The reaction mixtures were incubated at 40 °C for 60 min. After this time, the reactions were quenched by the addition of 40 μL of 1 N HCl, and the crude mixtures were diluted with 200 μL of acetonitrile and analyzed by LC/MS using the method described above. 

Retention times (min) for the derivatized (D-FDVA) amino acid standards under the reported conditions were as follows when Marfey’s analysis was performed for compounds **1**, **3**, and **4**: D-Ser: 4.97, L-Ser: 5.83, single adducts of D-Orn: 2.54 and 3.48, double adduct of D-Orn: 25.86, single adducts of L-Orn: 2.29 and 3.48, double adduct of L-Orn: 24.12 (Appendix A). Retention times (min) for the observed peaks in the HPLC trace of the D-FDVA-derivatized hydrolysis products of compounds were as follows: (**1**) D-Ser: 4.94, Gly: 7.86, β-Ala: 10.27, double adduct of L-Orn: 24.42, and double adduct of *N*α-methyl-L-Orn: 29.97 (Appendix A); (**3**) D-Ser: 4.93, Gly: 7.88, β-Ala: 10.29, double adduct of L-Orn: 24.40, and double adduct of *N*α-methyl-L-Orn: 29.98 (Appendix A); and (**4**) D-Ser: 4.91, Gly: 7.87, β-Ala: 10.27, double adduct of L-Orn: 24.47, and double adduct of *N*α-methyl-L-Orn: 29.86 (Appendix A).

### 3.6. Genome Sequencing and Biosynthetic Gene Clusters Prediction

High molecular-weight genomic DNA from strain CA-135719 was isolated following the protocol described in Practical Streptomyces Genetics. A Laboratory Manual [28] from 10 mL bacterial culture grown on ATCC-2 liquid medium (soluble starch 20 g/L, glucose 10 g/L, NZ Amine Type E 5 g/L, meat extract 3 g/L, peptone 5 g/L, yeast extract 5 g/L, calcium carbonate 1 g/L, pH 7) grown on an orbital shaker at 28 °C, 220 rpm, 70% relative humidity for 4 days.

The genome of the strain CA-135719 was sequenced, de novo assembled, and annotated by Macrogen (Seoul, Republic of Korea; http://www.macrogen.com, accessed on 21 February 2023), using a combined strategy of Illumina Novaseq 6000 and PacBio SequelII platforms [29,30]. The PacBio long reads were assembled with Raven (https://github.com/lbcb-sci/raven, accessed on 21 February 2023) [31], and then Illumina reads were mapped to the assembly for accurate genome sequence and error correction using Pilon (v1.21) [32]. After mapping, the consensus sequence was generated. The completeness of the genome was assessed with BUSCO (Benchmarking Universal Single-Copy Orthologous, v5.1.3) [33].

Then the genome was analyzed with antiSMASH v7.0.0beta1-67b538a9 [34] and PRISM v4 [35] to identify putative secondary metabolite biosynthetic gene clusters. BLAST (Basic Local Alignment Search Tool) [36] was also employed to predict the function of the genes.

### 3.7. Extraction of CA-135719 Cultures

Strain CA-135719 was cultured in fifteen Erlenmeyer flasks (500 mL containing 125 mL of culture medium) using the conditions described in Section 3.3. The flasks were extracted by addition of an equal volume of five different solvents, acetonitrile, n-butanol, ethyl acetate, methanol, and acetone, in triplicate. After addition of the solvent, flasks were shaken in a Kühner at 220 rpm for 2 h. The mixture was centrifuged (10 min, 8500 rpm) and filtered under vacuum, and the pellet was discarded. Organic solvents were evaporated under a nitrogen stream until half original volume to obtain the crude extracts. All crude extracts were analyzed by LC-HRMS. 

### 3.8. Madurastatin C1 Reaction with Acetone

Madurastatin C1 (0.4 mg) was incubated in an acetone:water mixture (1:1, 0.2 mL) for 2 h followed by LC-MS analysis [26]. 

### 3.9. Antiparasitic Activity

#### 3.9.1. Antiparasitic Activity on *T. cruzi*

Human osteosarcoma cell line U2OS (ATCC HTB-96) and rhesus monkey kidney epithelial cell LLM-MK2 (ATCC CCL-7) were purchased from the American Type Culture Collection (ATCC, Manassas, VA, USA). *T. cruzi* Y strain was maintained as tissue culture trypomastigotes (TCT) by infection to LLC-MK2 cell line. The U2OS human cell was cultivated in DMEM-high-glucose media supplemented with 10% heat-inactivated fetal bovine serum (FBS, Thermo Fisher Scientific, Waltham, MA, USA). To visualize amastigote replication, U2OS cells were seeded in a 384-well plate (4 × 10^3^ cells/well) and infected with 5 × 10^4^ TCT parasites in DMEM-low-glucose media supplemented with 2% heat-inactivated FBS as previously described [37]. The reference drug 400 μM of benznidazole (positive control), 0.5% DMSO (negative control), and the tested samples were treated on the day of assay and then incubated for 3 days at 37 °C, 5% CO_2_. After incubation was completed, the cells and parasites were fixed with 4% paraformaldehyde (PFA), and the DNA was stained with 5 μM DRAQ5 (Biostatus, Shepshed, Leicestershire, UK) followed by image acquisition using an automated confocal microscope (Operetta CLS, PerkinElmer, Waltham, MA, USA) with a 635 mm laser at 20× objective.

#### 3.9.2. Antiparasitic Activity on *L. donovani*

Human monocyte cell line THP-1 (ATCC TIB-202) was purchased from the American Type Culture Collection (ATCC). The assay for activity against *L. donovani* was performed as previously described [38]. *L. donovani* MHOM/ET/67/HU3 was cultivated as promastigotes at 28 °C in M199 (Sigma-Aldrich, St. Louis, MO, USA) with 40 mM HEPES, 0.1 mM adenine, 0.0001% biotin, and 4.62 mM NaHCO_3_, supplemented with 10% heat-inactivated FBS and incubated 6 days before infection to enrich the proportion of metacyclic promastigotes. THP-1 was cultivated in RPMI medium (Thermo Fisher Scientific) supplemented with 10% heat-inactivated FBS and differentiated with 50 ng/mL of phorbol 12-myristate 13-acetate (PMA, Sigma-Aldrich) before infection. Purified *L. donovani* parasites were seeded with 1:20 infection ratio in 384-well plates and incubated for 24 h, then followed by the treatment of 4 μM amphotericin B (Sigma-Aldrich) as positive control, 0.5% DMSO as negative control, and tested samples. After additional incubation for 3 days at 37 °C, 5% CO_2_, the cells and parasites were fixed with 4% PFA and stained with 5 μM DRAQ5. Fluorescent images were acquired from each well using an automated confocal microscope (Operetta CLS) with a 635 mm laser at 20× objective.

#### 3.9.3. Data Analysis

The acquired images were analyzed with Columbus software (v2.3) (PerkinElmer) to quantify cell numbers, parasites numbers, and infection ratio. Data were normalized by positive and negative controls. The activity of the tested samples was determined in terms of IC_50_ and CC_50_ calculated with XLfit software (v5) (IDBS, Boston, MA, USA). The quality of the assay data was assessed with Z’ factor, assay windows, and CV [37].

## 4. Conclusions

In conclusion, five madurastatin derivatives were isolated from an acetone extract of the strain CA-135719 of *Actinomadura* sp. Four of these compounds, containing 4-imidazolidinone rings, H2 (**2**), 33-*epi*-D1 (**3**), D1 (**4**), and D2 (**5**), were shown to be artifacts formed during the acetone extraction of the culture broths. Madurastatin C1 is therefore the only natural madurastatin produced by the strain *Actinomadura* sp. CA-135719 in significant titers. Based on the similarity between the *mad* and *mds* BGCs, we revised and corrected the absolute stereochemistry of the previously reported madurastatins D1 and D2 [4] (also isolated from an acetone extract), establishing their stereocenters’ absolute configuration as 9*R*, 23*S*, 26*S*, 33*S* and 9*R*, 23*S*, 26*S*, respectively, and based on our results and the biosynthetic evidence provided, we question the existence of *ent*-madurastatin C1 isolated in the same work. 

## Figures and Tables

**Figure 1 ijms-25-00301-f001:**
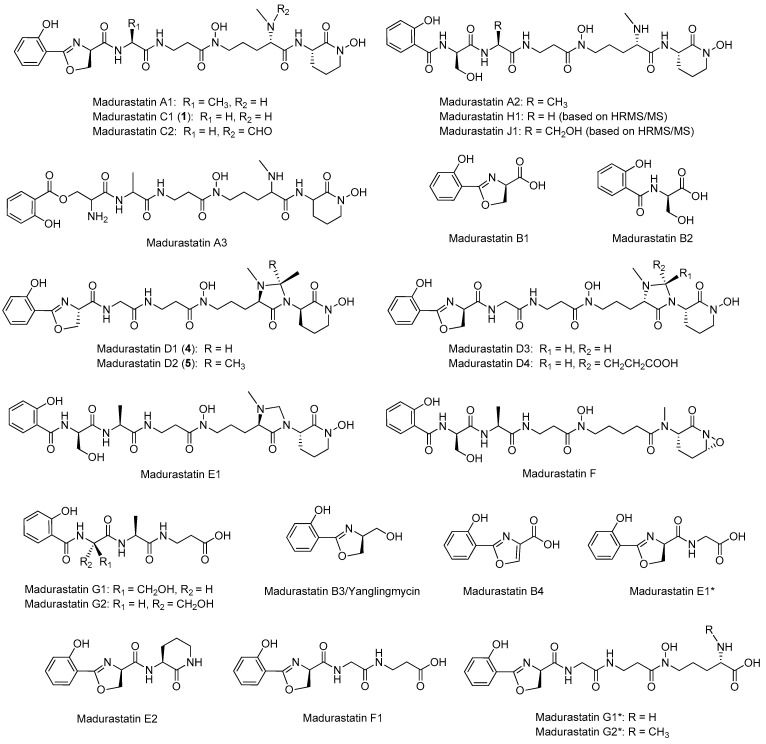
Chemical structures of known madurastatins. * Indicates those compounds whose names were reported earlier for compounds having different structures.

**Figure 2 ijms-25-00301-f002:**
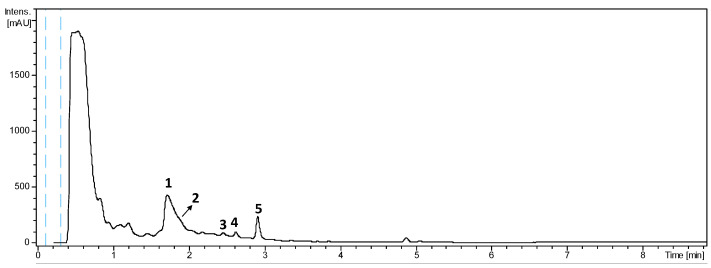
LC-HRMS profile of the acetone extract from *Actinomadura* sp. CA-135719.

**Figure 3 ijms-25-00301-f003:**
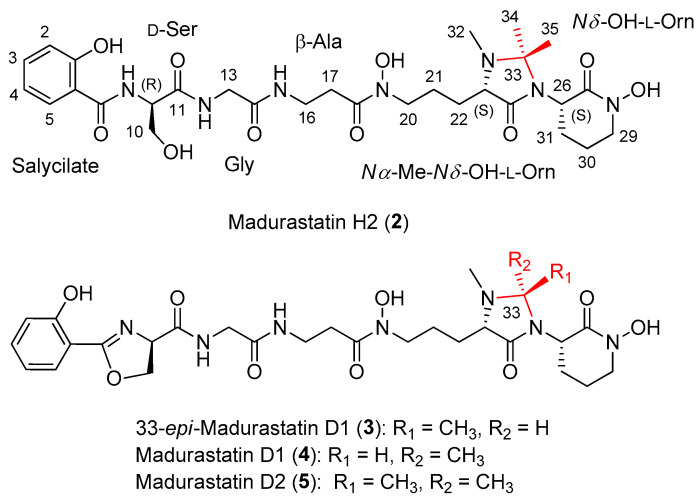
Structures of madurastatin H2 (**2**), 33-*epi*-madurastatin D1 (**3**), madurastatin D1 (**4**), and madurastatin D2 (**5**) isolated from the *Actinomadura* sp. CA-135719 culture.

**Figure 4 ijms-25-00301-f004:**
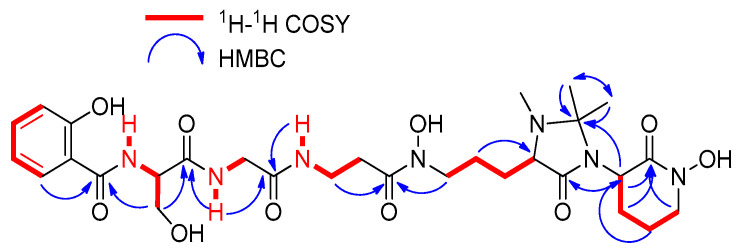
^1^H-^1^H COSY, and key HMBC correlations observed in the 2D NMR spectra of **2**.

**Figure 5 ijms-25-00301-f005:**
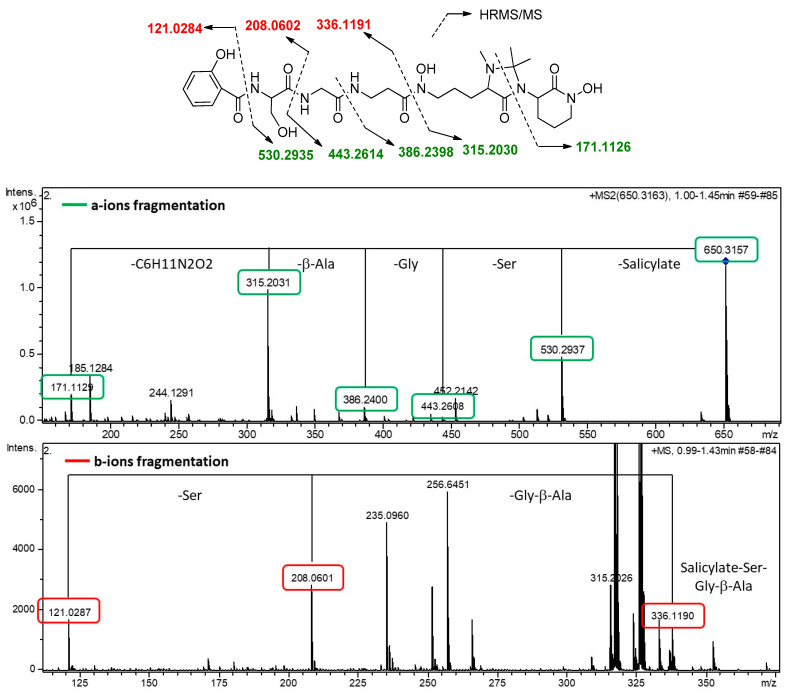
Fragments observed for **2** by MS^n^.

**Figure 6 ijms-25-00301-f006:**
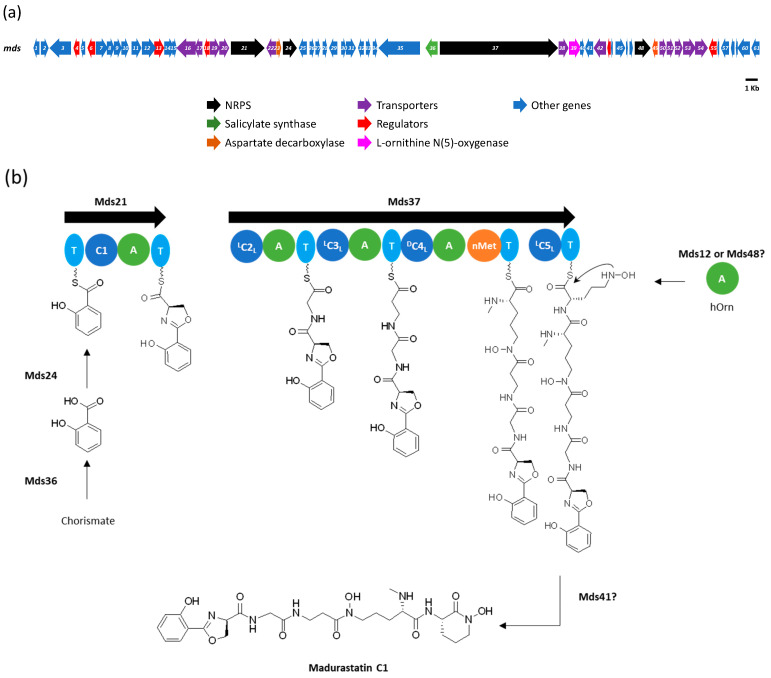
(**a**) Biosynthetic gene cluster for madurastatin (*mds*) from strain CA-135719. (**b**) Proposed assembly line for madurastatin C1 (**1**).

**Figure 7 ijms-25-00301-f007:**
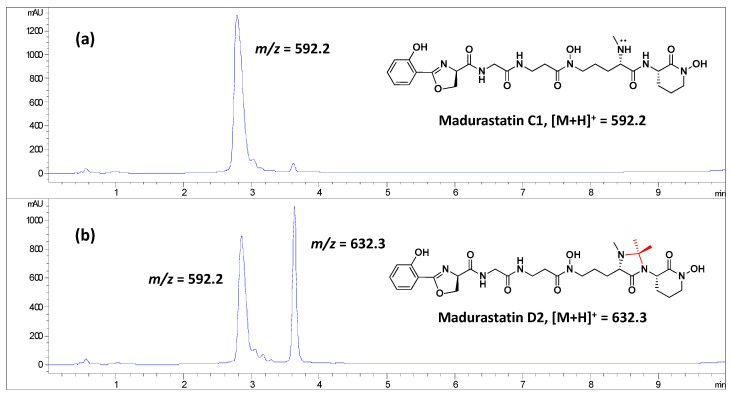
LC-MS profiles at 210 nm of (**a**) madurastatin C1, and (**b**) madurastatin C1 after incubation with acetone.

**Table 1 ijms-25-00301-t001:** NMR spectroscopic data (500 MHz, DMSO-*d*_6_) for madurastatin H2 (**2**).

Residue	Position	δ_C_, Type	δ_H_, Mult. (*J* in Hz)
Salicylate	1	160.1, C	-
	2	117.3, CH	6.99, d (8.3)
	3	136.0, CH	7.54, ddd (8.8, 7.4, 1.7)
	4	119.3, CH	6.95, ddd (8.8, 7.3, 0.8)
	5	131.0, CH	7.94, dd (7.9, 1.6)
	6	112.7, C	-
	7	167.8, C	-
Ser	8	NH	8.53, br s
	9	51.4, CH	4.37, br s
	10	63.2, CH_2_	4.65, m
			4.62, m
	11	166.0, C	-
Gly	12	NH	8.89, t (5.6)
	13	42.2, CH_2_	3.83, dd (16.5, 5.9)
			3.74, dd (16.5, 5.4)
	14	167.9, C	-
β-Ala	15	NH	8.02, t (5.4)
	16	34.7, CH_2_	3.25, m
	17	31.9, CH_2_	2.53, m
	18	170.9, C	-
*N*α-Methyl *N*δ-OH Orn	19	N	-
	20	47.1, CH_2_	3.50, m
	21	22.0, CH_2_	1.83, m
			1.57, m
	22	24.7, CH_2_	1.66, m
			1.58, m
	23	61.7, CH	3.58 *
	24	167.8, C	-
*N*δ-OH Orn	25	N	-
	26	52.2, CH	4.04, q (5.3)
	27	162.9, C	-
	28	N	-
	29	51.2, CH_2_	3.52, m
			3.44, m
	30	20.8, CH_2_	1.90, m
	31	25.3, CH_2_	2.37, m
			1.76, m
	32	33.0, CH_3_	2.56, m
	33	80.2, C	-
	34	23.3, CH_3_	1.50, br s
	35	20.3, CH_3_	1.32, br s

* Not clearly observed.

**Table 2 ijms-25-00301-t002:** NMR spectroscopic data (500 MHz, DMSO-*d*_6_) for 33-*epi*-madurastatin D1 (**3**) and madurastatin D1 (**4**).

		33-*epi*-Madurastatin D1 (3)		Madurastatin D1 (4)	
Residue	Position	δ_C_, Type	δ_H_, Mult. (*J* in Hz)	δ_C_, Type	δ_H_, Mult. (*J* in Hz)
Salicylate	2	116.3, CH	7.00, d (8.4)	116.3, CH	7.00, d (8.3)
	3	133.8, CH	7.47, ddd (8.4, 7.4, 1.3)	133.9, CH	7.47, ddd (8.3, 7.4, 1.3)
	4	118.8, CH	6.95, dd (8.1, 7.3)	118.8, CH	6.95, dd (8.2, 7.5)
	5	127.8, CH	7.64, dd (7.7, 1.3)	127.8, CH	7.64, dd (7.8, 1.3)
Ser	9	67.2, CH	5.01, dd (10.5, 7.8)	67.1, CH	5.01, dd (10.3, 7.8)
	10	69.2, CH_2_	4.65, dd (10.5, 8.5)	69.2, CH_2_	4.65, dd (10.3, 8.5)
			4.52, dd (8.8, 7.6)		4.52, dd (8.5, 7.7)
Gly	13	41.7, CH_2_	3.75, dd (16.5, 5.7)	41.9, CH_2_	375, dd (16.5, 5.9)
			3.66, dd (16.5, 5.4)		3.66, dd (16.5, 5.7)
β-Ala	16	34.4, CH_2_	3.26, m	34.3, CH_2_	3.25, m
	17	31.7, CH_2_	2.52, m	31.6, CH_2_	2.53, m
*N*α-Methyl *N*δ-OH Orn	20	46.9, CH_2_	3.46, m	47.1, CH_2_	3.46, m
	21	21.7, CH_2_	1.68, m	21.3, CH_2_	1.68, m
			1.46, m		1.44, m
	**22**	**24.1, CH_2_**	1.50, m	**26.3, CH_2_**	1.46, m
	**23**	**62.1, CH**	**3.14, m**	**64.6, CH**	**2.88, m**
*N*δ-OH Orn	**26**	51.2, CH	**4.13, q (5.9)**	51.5, CH	**4.27, m**
	29	50.9, CH_2_	3.52, m	50.8 CH_2_	3.54, m
			3.43, m		3.44, m
	30	20.6, CH_2_	1.91, m	20.5, CH_2_	1.87, m
	31	25.2, CH_2_	**2.17, m**	25.9, CH_2_	**1.98, m**
			1.77, m		1.84, m
	**32**	**33.4, CH_3_**	2.27, s	**37.6, CH_3_**	2.29, s
	**33**	73.5, CH	**4.42, m**	74.4, CH	**4.01, q (5.5)**
	**34**	**15.1, CH_3_**	**1.12, d (5.7)**	**19.4, CH_3_**	**1.21, d (5.4)**

Carbon chemical shifts obtained from HSQC spectrum. Chemical shifts affected by the different absolute configuration at C-33 are shown in bold.

**Table 3 ijms-25-00301-t003:** Ion extraction by LC-HRMS of the targeted compounds (**1**–**5**) using five different extraction solvents.

			Area Target Compound × 10^6^				
			Acetonitrile Extract	n-Butanol Extract	Ethyl Acetate Extract	Methanol Extract	Acetone Extract
Targeted Compounds	[M + H]^+^	RT (min)	Av. ± SD	Av. ± SD	Av. ± SD	Av. ± SD	Av. ± SD
**H1**	610.283 ± 0.005	0.8 ± 0.5	0.3 ± 0.1	0.2 ± 0.0	0.3 ± 0.0	0.2 ± 0.0	0.2 ± 0.0
**C1 (1)**	592.273 ± 0.005	1.7 ± 0.5	11.0 ± 1.5	8.6 ± 1.5	8.9 ± 0.6	9.6 ± 0.3	8.3 ± 0.7
**H2 (2)**	650.314 ± 0.005	1.9 ± 0.5	0.0 ± 0.0	0.0 ± 0.0	0.0 ± 0.0	0.0 ± 0.0	0.1 ± 0.0
**33-*epi* D1 (3)**	618.288 ± 0.005	2.3 ± 0.5	0.0 ± 0.0	0.0 ± 0.0	0.0 ± 0.0	0.0 ± 0.0	0.9 ± 0.3
**D1 (4)**	618.288 ± 0.005	2.6 ± 0.5	0.0 ± 0.0	0.0 ± 0.0	0.0 ± 0.0	0.0 ± 0.0	1.5 ± 0.2
**D2 (5)**	632.304 ± 0.005	2.9 ± 0.5	0.0 ± 0.0	0.0 ± 0.0	0.0 ± 0.0	0.0 ± 0.0	4.1 ± 0.3

The values of the areas are the average of three biological replicates as well as the standard deviation.

**Table 4 ijms-25-00301-t004:** Antiparasitic activity against *T. cruzi* and *L. donovani* of isolated compounds (**1**–**5**).

		Madurastatins				
		C1 (1)	H2 (2)	33-*epi* D1 (3)	D1 (4)	D2 (5)
*T. cruzi*	Inhibition norm. inf. ratio IC_50_ (µM)	>50	>50	>50	>50	>50
	Inhibition norm. para. num. IC_50_ (µM)	30.76	>50	46.08	27.23	8.93
	Cell ratio (%) CC_50_ (µM)	>50	>50	>50	46.77	24.74
*L. donovani*	Inhibition norm. inf. ratio IC_50_ (µM)	>50	>50	>50	>50	>50
	Inhibition norm. para. num. IC_50_ (µM)	>50	>50	>50	>50	>50
	Cell ratio (%) CC_50_ (µM)	>50	>50	>50	>50	>50

Inhibition norm. inf. ratio: Antiparasitic parameter normalized by negative (DMSO treated-infected) and positive (Benz. treated-infected) control that measures the number of host cells infected with parasite in their cytoplasmic region. Inhibition norm. para. num.: Antiparasitic parameter normalized by negative (DMSO treated-infected) and positive (Benz. treated-infected) control that measures the number of parasites in the cytoplasmic region of host cells. Cell ratio (%): cytotoxicity parameter that count the number of host cells.

## Data Availability

Data are contained within the article and Appendix A.

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
