# Peer review of "Madurastatins with Imidazolidinone Rings: Natural Products or Side-Reaction Products from Extraction Solvents?"

_ijms, 2023, doi:10.3390/ijms25010301_

Round 1

Reviewer 1 Report

Comments and Suggestions for Authors

This manuscript details some studies on the madurastatins, a group of modified pentapeptides, as potential treatments for the neglected tropical diseases caused by the parasites T. cruzi and L. donovani. Two new madurastatins (H2 and 33-epi-D1) are reported, and the structure of madurastatin D1 and D2 were corrected. Additionally, madurastatins containing imidazolidinone rings are proved to be artifacts 22 originating during acetone extraction of the bacterial cultures. Overall I think the results are interesting and useful to chemists working on these natural products. The experimental section and supplementary information appear to be complete.

I did find a few minor issues with Figure 6. Specifically:

-The authors show the arrow starting from the OH of the hydroxylamine to form the lactam, I think the arrow should begin at the NH. I'm also not clear on the arrow starting from the hOrn domain, is this supposed to remove the H from the NH?

-Also, I'm not sure what the arrow that is initiated at the nMet domain is signifying, is this an error?

Author Response

Dear reviewer,

Thank you for your contribution. Please find below the answers to your comments.

Reviewer 1

This manuscript details some studies on the madurastatins, a group of modified pentapeptides, as potential treatments for the neglected tropical diseases caused by the parasites T. cruzi and L. donovani. Two new madurastatins (H2 and 33-epi-D1) are reported, and the structure of madurastatin D1 and D2 were corrected. Additionally, madurastatins containing imidazolidinone rings are proved to be artifacts 22 originating during acetone extraction of the bacterial cultures. Overall, I think the results are interesting and useful to chemists working on these natural products. The experimental section and supplementary information appear to be complete.

I did find a few minor issues with Figure 6. Specifically:

-The authors show the arrow starting from the OH of the hydroxylamine to form the lactam, I think the arrow should begin at the NH. Thank you for the correction. Certainly, the arrow was displaced and should begin at the NH. We have corrected this in a new version of Figure 6.

I'm also not clear on the arrow starting from the hOrn domain, is this supposed to remove the H from the NH? As we explain in the text, a trans-acting A-domain (Mds12 or Mds48) may activate the Nδ-hydroxy-L-ornithine incorporated in the last module of Mds37, and the arrow represents the incorporation of this residue into the NRPS. To clarify this step, we have changed the arrow in the new version of Figure 6.

-Also, I'm not sure what the arrow that is initiated at the nMet domain is signifying, is this an error? This arrow represented the N-methylation of the incorporated ornithine residue, but to avoid any confusion, we have deleted this arrow in the new Figure 6.

Best regards, Mercedes.

Reviewer 2 Report

Comments and Suggestions for Authors

In this manuscript, authors investigated the chemical constituents of the acetone extracts from a culture of Actinomadura sp. CA-135719. As a result, five madurastatin derivatives were obtained. Interestingly, the imidazolidinone rings in the structures of madurastatins H2, 33-epi-D1, D1, and D2 have been shown to be artifacts formed during the acetone extraction of the culture broths. The absolute configurations of these isolates were proposed using Marfey′s and bioinformatic BGC analyses, leading to a revision of the absolute configurations of madurastatins D1 and D2. In the bioassay, madurastatins exhibited moderate to low antiparasitic activity against Trypanosoma cruzi. All these findings were important. This is a nice manuscript, which could be attractive for readers.

1. Interestingly, it was reported the existence of ent-madurastatin C1, proposed by Yan et al. previously, was not reliable. Authors provided many evidence to show the irrationality of the configuration of this compound. Did you find the case that the sign of specific rotation value became the opposite when measured with a metal-chelated species of the molecule? Since madurastatin C1 was obtained in this study, perhaps authors could measure a metal chelate.

2. It is better to move Figure 5 forward and put below the first paragraph of the subsection ‘2.2. Extraction, Isolation and Structural Elucidation’, which could make the readers easily find the numberings of the reported compounds.

3. As shown in Figure 4, the detected HRESIMS data for compound 2 was m/z 650.3157 [M + H]+, which was not consistent with 650.3149 (P3L83). Please provide the HRMS spectra for the new compounds 2 and 3 in the Supplementary Information.

4. Poor solution for the third chromatograph in Figure S17.

Other revisions:

1. Abstract: It is better to give a full name ‘biosynthetic gene cluster’ for the abbreviation ‘BGC’.

2. P10L260: ‘table 4’ → ‘Table 4’

3. Please use the Italic font for ‘T. cruzi’ and ‘L. donovani’ in the caption of Table 4.

Author Response

Dear reviewer,

Thank you for your contribution. Please find below the answers to your comments.

Reviewer 2

In this manuscript, authors investigated the chemical constituents of the acetone extracts from a culture of Actinomadura sp. CA-135719. As a result, five madurastatin derivatives were obtained. Interestingly, the imidazolidinone rings in the structures of madurastatins H2, 33-epi-D1, D1, and D2 have been shown to be artifacts formed during the acetone extraction of the culture broths. The absolute configurations of these isolates were proposed using Marfey′s and bioinformatic BGC analyses, leading to a revision of the absolute configurations of madurastatins D1 and D2. In the bioassay, madurastatins exhibited moderate to low antiparasitic activity against Trypanosoma cruzi. All these findings were important. This is a nice manuscript, which could be attractive for readers.

  1. Interestingly, it was reported the existence of ent-madurastatin C1, proposed by Yan et al. previously, was not reliable. Authors provided many evidence to show the irrationality of the configuration of this compound. Did you find the case that the sign of specific rotation value became the opposite when measured with a metal-chelated species of the molecule? Since madurastatin C1 was obtained in this study, perhaps authors could measure a metal chelate. The fact that the specific rotation value shows an opposite sign as a consequence of measuring a metal-quelated madurastatin is just a possible explanation to this finding, since madurastatins are metal chelating agents. However, other explanations are also possible and a screening of the specific rotation of the compound without knowing the chelating metal is envisaged as a challenging task. We believe that we have provided enough evidence to demonstrate that ent-madurastatin C1 does not exist based on the fact that BGC analysis provides almost identical results in our strain and the strain described by Yan et al. and in the rigor of our chemical analysis compared to them.
  2. It is better to move Figure 5forward and put below the first paragraph of the subsection ‘2.2. Extraction, Isolation and Structural Elucidation’, which could make the readers easily find the numberings of the reported compounds. We have moved Figure 5, so now is Figure 3, thus, it will be easy for the readers to find the numbering of the reported compounds. We have reorganized the text and numbering of the rest of the figures accordingly.
  3. As shown in Figure 4, the detected HRESIMS data for compound 2was m/z 650.3157 [M + H]+, which was not consistent with 650.3149 (P3L83). Please provide the HRMS spectra for the new compounds 2and 3 in the Supplementary Information. The difference between the values of m/z comes from two different analyses of the same sample. Whereas Figure 4 shows the MS/MS spectrum of the sample, the m/z value was obtained from the HRMS spectrum (which is included now in the Supplementary Information). The difference between both m/z values is less than 2 ppm, which is within the range acceptable for two different injections of the same sample.
  4. Poor solution for the third chromatograph in Figure S17. It is true, the signal of the EIC is very poor for the [M+H]+ of the double adduct ornithine-D-FDVA, however, the sodium adduct can be observed and it is similar to the sodium adducts observed for compound 1 (Figure S9) and compound 3 (Figure S13), so we can conclude that ornithine also has the L-configuration.

Other revisions:

  1. Abstract: It is better to give a full name ‘biosynthetic gene cluster’ for the abbreviation ‘BGC’. Certainly, we have indicated in the abstract the full name.
  2. P10L260: ‘table 4’ → ‘Table 4’. Corrected.
  3. Please use the Italic font for ‘T. cruzi’ and ‘L. donovani’ in the caption of Table 4. Corrected.

Best regards, Mercedes.
